# Impact of recent climate extremes on mosquito-borne disease transmission in Kenya

Cameron Nosrat[1]*, Jonathan Altamirano[2], Assaf Anyamba[3], Jamie M. Caldwell[4], Richard Damoah[5], Francis Mutuku[6], Bryson Ndenga[7], A. Desiree LaBeaud[2]

**1** Program in Human Biology, Stanford University, Stanford, California, United States of America, **2** Department of Pediatrics, Stanford University School of Medicine, Stanford, California, United States of America, **3** Universities Space Research Association & NASA Goddard Space Flight Center, Greenbelt, Maryland, United States of America, **4** Department of Biology, Stanford University, Stanford, California, United States of America, **5** Morgan State University & NASA Goddard Space Flight Center, Greenbelt, Maryland, United States of America, **6** Technical University of Mombasa, Mombasa, Kenya, **7** Centre for Global Health Research, Kenya Medical Research Institute, Kisumu, Kenya

* cnosrat@stanford.edu

**Data Availability Statement:** Data are available from the Stanford Repository at the following URL: https://purl.stanford.edu/rz262rz1347.

## Abstract

Climate change and variability influence temperature and rainfall, which impact vector abundance and the dynamics of vector-borne disease transmission. Climate change is projected to increase the frequency and intensity of extreme climate events. Mosquito-borne diseases, such as dengue fever, are primarily transmitted by *Aedes aegypti* mosquitoes. Freshwater availability and temperature affect dengue vector populations via a variety of biological processes and thus influence the ability of mosquitoes to effectively transmit disease. However, the effect of droughts, floods, heat waves, and cold waves is not well understood. Using vector, climate, and dengue disease data collected between 2013 and 2019 in Kenya, this retrospective cohort study aims to elucidate the impact of extreme rainfall and temperature on mosquito abundance and the risk of arboviral infections. To define extreme periods of rainfall and land surface temperature (LST), we calculated monthly anomalies as deviations from long-term means (1983–2019 for rainfall, 2000–2019 for LST) across four study locations in Kenya. We classified extreme climate events as the upper and lower 10% of these calculated LST or rainfall deviations. Monthly *Ae. aegypti* abundance was recorded in Kenya using four trapping methods. Blood samples were also collected from children with febrile illness presenting to four field sites and tested for dengue virus using an IgG enzyme-linked immunosorbent assay (ELISA) and polymerase chain reaction (PCR). We found that mosquito eggs and adults were significantly more abundant one month following an abnormally wet month. The relationship between mosquito abundance and dengue risk follows a non-linear association. Our findings suggest that early warnings and targeted interventions during periods of abnormal rainfall and temperature, especially flooding, can potentially contribute to reductions in risk of viral transmission.

**Funding:** This research was supported by National Institutes of Health (NIH) grant R01AI102918 (ADL). The Stanford REDCap platform (http://redcap.stanford.edu) is operated by Stanford Medicine Research IT team. The REDCap platform services at Stanford are subsidized by a) Stanford School of Medicine Research Office, and b) the National Center for Research Resources and the National Center for Advancing Translational Sciences, National Institutes of Health, through grant UL1 TR001085‡. JMC was supported by a Stanford Woods Institute for the Environment - Environmental Ventures Program grant. The funders had no role in the collection, analysis, or reporting of the data.

**Competing interests:** The authors have declared that no competing interests exist.

## Author summary

Dengue is a rapidly spreading mosquito-borne disease transmitted primarily by *Aedes aegypti* mosquitoes. As climate change leads to extremes in rainfall and temperature, the abundance and populations of these vectors will be affected, thus influencing transmission of dengue. Using satellite-derived climate data for Kenya, we classified months that experienced highly abnormal rainfall and temperature as extreme climate events (floods, droughts, heat waves, or cold waves). We compared the average monthly *Ae. aegypti* abundance and confirmed dengue counts following extreme climate months using lag periods of one month and two months, respectively. This study utilized several statistical models to account for differences among study sites and time. Floods resulted in significantly increased egg and adult abundance. Our results contributed to a better understanding of the effect of climate variability and change on dengue. As suggested by our observed increase in vector counts yet a relatively unchanged dengue infection risk, human behavior can help reduce viral transmission. Targeted interventions should be focused on both reducing vector populations and limiting human-vector contact, especially during these climate anomalies.

## Introduction

Climate change's influence on temperature and rainfall can dramatically impact vector abundance and thus the dynamics of vector-borne disease transmission. Scientific evidence suggests that as global climate change continues to intensify, so will the frequency of extreme climate events, including floods, drought, heat waves, and cold waves [1–4]. Extreme climate events result from both natural internal climate variability and climate warming [5]. Climate variability arises spontaneously within the climate system even in the absence of climate forcings [5]. With regards to climate warming, sea-surface temperature increases of 1–2˚ Celsius can result in greater trapped greenhouse gas molecules and energy flux in the lower atmosphere, resulting in stronger, more unpredictable weather patterns [1]. Water availability during extreme climate events, including droughts and floods, can have important implications for mosquito-borne disease transmission, as pools of water provide breeding sites for infectious disease vectors. Similarly, local temperatures can alter mosquito dynamics, including the development of immature mosquitoes, and rates of reproduction and biting [6]. Most notably, transmission of dengue fever, a prevalent arthropod-borne disease in Kenya that is the focus of this study, can be affected by variability in temperature and rainfall [6, 7].

Dengue is the most common and the fastest spreading vector-borne disease globally, resulting in close to half of the world's population living in areas at risk for dengue virus (DENV) transmission [8]. DENV is a flavivirus with four distinct serotypes (1–4) found mostly in tropical and sub-tropical regions of the world [8]. It is transmitted by *Aedes aegypti* and *Aedes albopictus* mosquito vectors [9]. In 2019, the largest number of DENV infections were reported globally; annual infection counts have increased 15-fold over the past two decades, resulting in large dengue epidemics [8]. DENV is transmitted by mosquitoes throughout daylight hours and human populations living in close contact with mosquito vector breeding sites are at risk for DENV infection. The prevalence of vector breeding sites, both natural and artificial, in combination with ambient temperatures can influence vector abundance, vector growth, and infectious disease transmission. However, the effect of extreme climate events, including floods, droughts, heat waves, and cold waves on mosquito-borne disease transmission is not well understood.

Previous studies indicate that accumulated rainfall increases vector habitats, but floods and excessive rainfall flush breeding sites, thus diminishing vector populations of several mosquito species, including *Ae. aegypti* [10–13]. For example, in Singapore between 2014–2015, researchers observed excessive rainfall flushed *Ae. aegypti* breeding sites and decreased the risk of dengue outbreaks six weeks following rainfall [11]. Several epidemiological studies also support strong associations between accumulated rainfall and higher vector abundance 4+ weeks later [14–16]. For example, *Aedes albopictus*, a closely related species of *Ae. aegypti* and vector for dengue virus and chikungunya virus, was observed to be positively associated with accumulated rainfall at a lag of four weeks in southern France. This ultimately contributed to an increased risk of chikungunya transmission in France in 2014 [17]. Such a relationship held true for dengue as well, as researchers in Jakarta and Bali observed the number of dengue cases to increase with higher monthly mean rainfall up to 16.2 mm over the past decade [18]. As such, the relationship between rainfall and dengue transmission is not clearly delineated.

While a common view is that reductions in water availability removes vector breeding sites and diminish mosquito populations, drought conditions seem favorable for certain mosquito species [19, 20]. There are several modes through which drought is believed to promote greater vector abundance, according to a UK literature review; the primary mechanism through which *Ae. aegypti* abundance is promoted is through increased storage of water, which increases the availability of aquatic habitats for mosquitoes [20]. Such droughts contributed to several mosquito-borne diseases outbreaks, including a chikungunya virus outbreak in coastal Kenya between 2004–2005 [20, 21].

Temperature can affect many mosquito biological processes (e.g., reproduction, biting rate, development rate, etc.), thus influencing the prevalence of mosquitoes and the extent of disease spread [22]. Researchers have identified that the thermal response curve for *Ae. aegypti* transmission of dengue virus peaks at 29°C, which is higher than optimal transmission temperatures for other vector species [23]. However, much remains unknown about whether such laboratory-measured temperature thresholds lead to defined thresholds in reality. A study examining the effect of climatic factors on dengue transmission between Bali and Australia observed the number of dengue cases in Bali to increase with increasing mean temperature [18]. Heat waves have also been associated with increased dengue transmission; between December 2010-February 2011, seasonal land surface temperatures were 5–20°C above normal, and these above-normal temperatures were associated with the first known large-scale outbreak of dengue fever in East Africa [24]. We hope to better contextualize previous temperature thresholds and trends associated with DENV transmission using field data in Kenya.

Kenya is an ideal study site to better understand the relationship between climate variability, vector abundance, and mosquito-borne disease transmission due to the endemicity of mosquito-borne diseases, like dengue fever, and the region's interannual variable climate due to the El Niño Southern Oscillation [25, 26]. Due to the generally low levels of endemic DENV transmission in the region, more than a handful of cases are usually associated with small dengue outbreaks. Using existing vector (January 2014 –September 2018), climate (November 2013 –February 2019), and disease (January 2014 –February 2019) data systematically collected over the past six years at four sites in Kenya (Fig 1), this study aims to identify how periods of extreme rainfall and temperature affect mosquito abundance and the risk of dengue infection in a cohort of Kenyan children.

## Methods

### Ethics statement

The study protocol was approved by the Stanford University Institutional Review Board (Protocol ID #31488) and the Kenya Medical Research Institute (KEMRI) National Scientific and

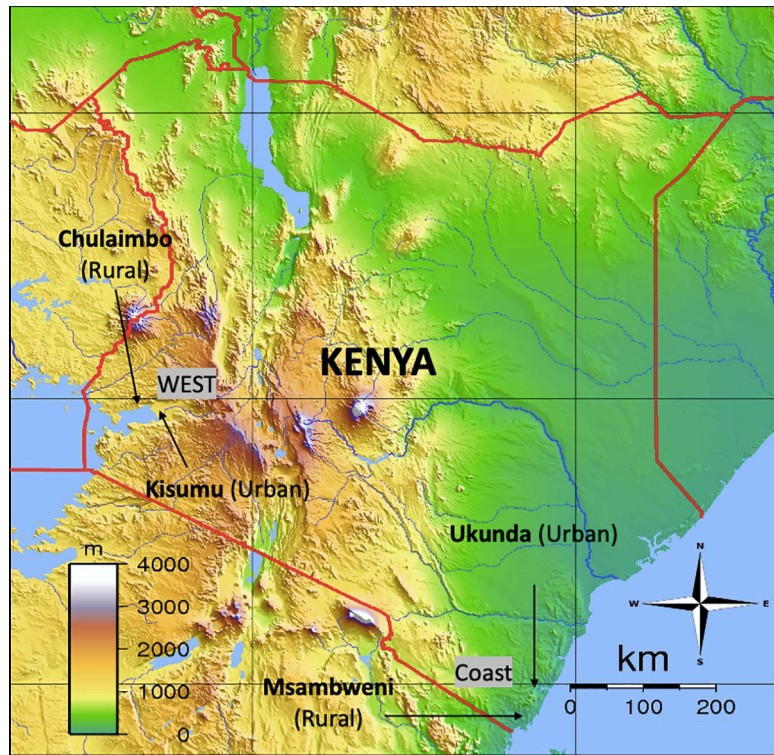

**Fig 1. Map of Study Sites in Kenya.** Image reused and altered from public domain (vidiani.com).

Ethical Review Committee (SSC # 2611). Meetings were held at all four sites with local government administrators (village elders, chiefs, and assistant chiefs) and with the local residents in each sub-location to introduce the research study and staff to the public. Written consent was obtained from all participants to collect blood samples and from household heads to sample mosquitoes within their houses and their compounds. Parents and guardians provided written consent on behalf of children and children >7 years of age also provided written assent. Mature minors provided written consent.

## Study sites

This study took place at two sites in western Kenya, Chulaimbo (rural) and Kisumu (urban), and two sites in coastal Kenya, Ukunda (urban) and Msambweni (rural) (Fig 1). The study sites vary in DENV burden, climate, geography, population size, and urbanization.

## Climate anomalies

Monthly mean land surface temperatures were extracted from the National Aeronautics and Space Administration's global monthly MOD11C3 version 6 data set derived from MODIS Terra. LST data are available at a spatial resolution 0.05˚ x 0.05˚ (≈5.5 km), thus allowing us to compare more recent monthly LSTs to long-term means for a 30 km x 30 km grid centered on each study location (S1 Appendix). The monthly long-term means were calculated using the 2003–2018 base period; that is, the long-term mean for each month was calculated by averaging the monthly mean LST for each month between 2003–2018. The study period was included in the base period, as the inclusion of all available observations allows for a more accurate characterization of the study locations' climate [27]. Average monthly temperatures between 2013–2019 were similarly gridded at 30km x 30km centered on each study location.

In order to assess rainfall variability for the study sites in Kenya, we used the daily African Rainfall Climatology Version 2 (ARC2) dataset from the archives of the National Oceanic and Atmospheric Administration (NOAA)–Climate Prediction Center (CPC). The rainfall estimates are gridded at 0.1˚ x 0.1˚ (≈11 km) spatial resolution operationally produced by a combination of rain gauge measurements and METEOSTAT satellites, thus providing rainfall estimates from 1982 to the present over Africa. Monthly long-term means were calculated for 30km x 30km grids for two counties, Kwale and Kisumu, using the 1982–2019 base period. As referenced in the WMO's Guidelines on the Calculation of Climate Normals, the inclusion of all available observations allows for a more accurate characterization of the study locations' climate, especially when fewer than 30 years of data are available [27]; as a result, the reference periods for LST and rainfall differ for this study. To calculate absolute monthly rainfall between 2013–2019, we aggregated data to a 30 km x 30 km grid centered on each study location. Absolute monthly rainfall for the coastal sites of Msambweni and Ukunda were compared to the long-term monthly mean rainfall for their county, Kwale. Monthly rainfall for the western sites of Kisumu and Chulaimbo were compared to the long-term monthly mean rainfall for their county, Kisumu.

Monthly climate anomalies ($X_a$) were defined as the difference between absolute monthly measurements (X) and long-term monthly means ($X_u$) for rainfall and land surface temperature:

$$X_a = X - X_u \qquad (1)$$

Extreme climate events were defined as anomalies greater than the 90th percentile and lower than the 10th percentile of anomalies (Fig 2). Since this study is concerned with how climate anomalies resulting from climate variability influence dengue transmission, classification of climate events does not consider absolute values of rainfall and LST but rather deviations from what is typically expected. More specifically, a flood is categorized as a positive rainfall deviation (i.e., above the 90% rainfall threshold), and a drought is categorized as a negative rainfall deviation (i.e., below the 10% rainfall threshold). A heat wave is categorized as a positive LST deviation (i.e., above the 90% LST threshold), and a cold wave is categorized as a

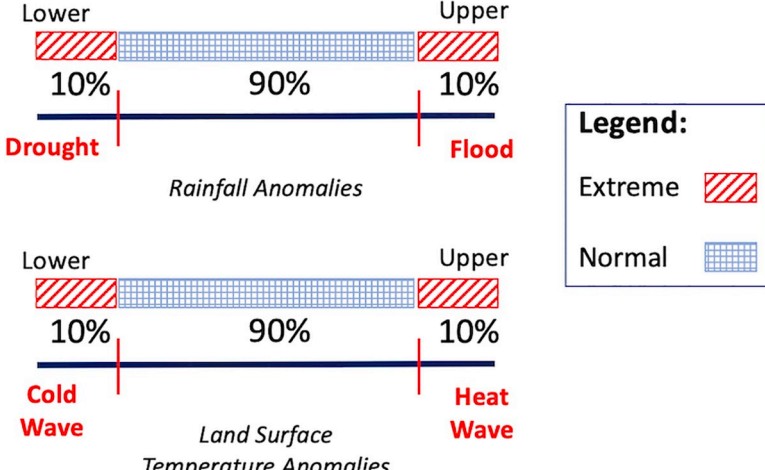

**Fig 2. Defining Extreme Climate Anomalies.** Extreme climate anomalies were defined as the upper and lower 10% of all anomalies (difference compared to long-term mean). For rainfall, upper 10% is designated as a "flood"; lower 10% is designated as a "drought." For LST, upper 10% is designated as a "heat wave"; lower 10% is designated as a "cold wave."

negative LST deviation (i.e., below the 10% rainfall threshold). Such a method of defining extreme climate periods allows for sufficient observations for analysis.

## Vector abundance

Mosquitoes of different life stages were sampled and classified by trapping method, date of collection, species and sex. This study is concerned with the abundance of *Aedes aegypti* and Aedes spp. (i.e. *Aedes ochraceus, Aedes fulgens Aedes pembaensis*), as they are primarily responsible for the transmission of DENV in Kenya. While traditionally an endophilic species, these mosquitoes have been observed to be primarily exophilic daytime feeders with peak biting periods early in the morning and in the evening before dusk in our study sites [28]. Vector abundance was recorded using various sampling methods to recover mosquito life stages, including ovitrap (eggs), pupal sampling (pupae), Prokopack aspiration (adults), and Biogents-sentinal (BG) trapping (adults) (S2 Appendix). The use of various collection methods allows for a more representative and accurate estimate of vector abundance, as each method allows for the collection of different mosquito life stages.

## Disease transmission

Dengue (DENV) incidence was assessed based on blood samples collected from children with acute febrile illness presenting to one of the four study sites (Mbaka Oromo Health Centre in Chulaimbo, Obama Children's Hospital in Kisumu, Msambweni District Hospital in Msambweni, and Ukunda/Diani Health Center in Ukunda) (Fig 1). The study population consisted of 7,653 children less than 18 years of age (median = 5 years [1 year, 15 years]). Unlike other places around the world, children in Kenya spend a lot of time outside during the day, which is when *Ae. aegypti* are actively biting. Blood samples were tested on site in Kenya by Ministry of Health collaborators at Msambweni District Hospital for coastal Kenya sites and KEMRI Kisian Field Station for western Kenya sites; samples were also tested at Stanford University. Cases of DENV were defined as a positive by polymerase chain reaction (PCR) and/or IgG-positive enzyme linked immunosorbent assay (ELISA)–children were considered dengue positive at the initial visit if viremia was found in the blood at the initial visit (e.g. by PCR) and if they seroconverted based on the follow-up visit (e.g. by PCR and IgG-ELISA). If a child presented PCR negative and already had antibodies at the initial visit, they were not included in the totals because this indicates that they had dengue previously at some point in their life and we would not be able to distinguish whether an infection occurred in the last few weeks.

## Statistical analyses

We conducted bivariate analyses, including Kruskal-Wallis tests by ranks and Wilcoxon rank sum tests, to investigate whether average *Ae. aegypti* abundance one month following each extreme climate event was significantly different compared to following "normal" climate (S3 Appendix). A one-month lag period was used between mosquito abundance and meteorological variables, as has been traditionally been supported in the literature and also expected by the cycle of infection [29]. An identical analysis was done for DENV infection counts at a lag of two months following the anomaly (S3 Appendix). While there have been observed time lags of 1–3 months between dengue incidence and meteorological variables, we made use of a two month lag period; rainfall and temperature have been observed to have the most prominent effects on dengue incidence at a lag period of two months [30, 31].

To test the effect of magnitude of rainfall and temperature anomalies on *Ae. aegypti* abundance after one month, we developed generalized log-linear mixed models for each mosquito life history stage (i.e., trapping method). The predictor variables for the model include rainfall

anomalies and LST anomalies while site, month, and year are included as random effects. We also tested for a possible interaction between rainfall anomalies and LST anomalies. Other variables, such as monthly accumulated rainfall, monthly LST, monthly ambient temperature, and humidity were excluded from the model because of their collinear relationships with rainfall anomalies and LST anomalies.

In addition to a generalized log-linear mixed model, we conducted a multinomial logistic regression model using the classification of rainfall anomalies and LST anomalies as our primary independent variables in explaining expected vector abundance classification the following month. Vector abundance was classified as low, intermediate, and high for each trapping method. Due to the non-normal distributions of the outcome, cutoffs for grouping were established non-uniformly across the trapping methods (S3 Table). As such, we were interested in observing how our categorization of climate anomalies helps predict vector counts. We calculated adjusted odds ratios including fixed effects for site and month, allowing us to consider site and time-endogenous variation.

Similarly, we tested the effect of rainfall and temperature anomalies on the number of monthly DENV infections using a two-month lag. We developed a binary logistic regression model, with the first outcome defined as <7 confirmed DENV infection in a month and the second outcome defined as 7+ confirmed DENV infections. 7+ confirmed DENV infections represent the upper 10% of monthly cases and can thus be defined as "higher" than normal in a region with low levels of DENV transmission. The model controlled for seasonality and regional differences by considering month, year, and site.

A second binary regression model tested the effect of rainfall and LST anomaly categorization on dengue transmission, again accounting for month, year, and site. Missing data was excluded from all models (S4 Table).

Descriptive and inferential analyses were conducted using the statistical software R (version 1.1.383, 2017, Boston, USA).

## Results

### Climate anomalies

Establishing defined thresholds for our categorization of extreme climate events allows us to compare related climatic variables of interest between types of extreme climate. Calculations and categorization of extreme climate events were based on previous guidelines [32]. A comparison between these groups suggests appropriate categorization of anomalies (S1 and S2 Tables). However, such categorizations of extreme climate events are unevenly distributed among the study sites (Tables 1 and 2).

**Table 1. Distribution of Rainfall Categorization.**

| Variable | Flood (N = 26) | Drought (N = 26) | Normal Rainfall (N = 204) | p-value |
|---|---|---|---|---|
| **Site**, N (%) | | | | **<0.001** |
| Chulaimbo | 3 (11.5) | 12 (46.2) | 49 (24.0) | |
| Kisumu | 0 (0.0) | 11 (42.3) | 53 (26.0) | |
| Msambweni | 14 (53.8) | 2 (7.7) | 48 (23.5) | |
| Ukunda | 9 (34.6) | 1 (3.8) | 54 (26.5) | |

Distribution of "flood", "drought", and "normal rainfall" categorizations across Kenyan study sites. P-values indicate significance values from Kruskal-Wallis Rank Sum tests among three groups. Dark grey sites represent those in western Kenya, which are drought-prone, while light grey sites represent those in coastal Kenya, which are flood-prone.

**Table 2. Distribution of LST Categorization.**

| Variable | Heat Wave (N = 26) | Cold Wave (N = 26) | Normal LST (N = 204) | p-value |
|---|---|---|---|---|
| **Site**, N (%) | | | | **<0.001** |
| Chulaimbo | 7 (26.9) | 6 (23.1) | 51 (25.0) | |
| Kisumu | 15 (57.7) | 15 (57.7) | 34 (16.7) | |
| Msambweni | 2 (7.7) | 1 (3.8) | 61 (29.9) | |
| Ukunda | 2 (7.7) | 4 (15.4) | 58 (28.4) | |

Distribution of "heat wave", "cold wave", and "normal LST" categorizations across Kenyan study sites. P-values indicate significance values from Kruskal-Wallis Rank Sum tests among three groups. Dark grey sites represent those in western Kenya, while light grey sites represent those in coastal Kenya.

## Vector abundance

***Ae. aegypti* eggs.** In our generalized log-linear model, rainfall anomalies were positively associated with egg abundance (p = 0.017); thus, for every ten-millimeter increase in rainfall anomalies (i.e., more severe floods) in any given month, a 2% increase in *Ae. aegypti* egg abundance would be expected, when site, month, and year, are included as random effects (Table 3). The effect of LST anomalies, and the interaction between rainfall anomalies and LST anomalies were insignificant.

Floods significantly increased the odds of egg abundance being classified as "high" when site, month, year, and LST classification were held constant in our multinomial logistic model (OR = 13.8 [6.5, 29.3], p < 0.001); flood classification increased the odds of "high" mosquito egg abundance by 1280% (Table 4). Drought, on the other hand, decreased the odds of egg abundance to be classified as "high" (OR = 0.70 [0.54, 0.90], p = 0.01) (Table 4). Heat waves decreased the odds of both "low" egg abundance (OR = 0.32 [0.23, 0.44], p < 0.001) and "high" egg abundance (OR = 0.22 [0.20, 0.23], p < 0.001), meaning that excessively increased LST anomalies (i.e., heat waves) would be expected to result in intermediate vector counts the following month. Cold waves decreased the odds of "low" egg abundance the following month (OR = 0.25 [0.20, 0.32], p < 0.001) (Table 4).

**Table 3. Generalized Log-Linear Model–Anomaly Severity and *Ae. aegypti* Abundance.**

| Trapping Method | Variable | Coefficient | Standard Error | t-Value | OR (95% CI) | p-Value |
|---|---|---|---|---|---|---|
| Ovitrap | **Rainfall Anomaly** | 2E-3 | 7E-4 | 2.39 | 1.002 (1.000, 1.003) | **0.017** |
| | **LST Anomaly** | 0.035 | 0.23 | 1.52 | 1.036 (0.989, 1.085) | 0.13 |
| Pupal Trapping | **Rainfall Anomaly** | 3E-4 | 7E-4 | 0.48 | 1.000 (0.999, 1.001) | 0.63 |
| | **LST Anomaly** | -0.03 | 0.03 | -1.20 | 0.970 (0.922, 1.020) | 0.23 |
| Prokopack | **Rainfall Anomaly** | -3E-4 | 8E-4 | -0.41 | 1.000 (0.999, 1.001) | 0.68 |
| | **LST Anomaly** | -0.091 | 0.03 | -3.41 | 0.913 (0867, 0.962) | **<0.001** |
| BG-Trap | **Rainfall Anomaly** | -5E-4 | 9E-4 | -0.582 | 0.999 (0.981, 1.017) | 0.56 |
| | **LST Anomaly** | -.01 | 0.03 | -0.427 | 0.988 (0.939, 1.041) | 0.67 |

Generalized log-linear mixed model with site, month, and year as random effects. Test of interaction: P value of > 0.05 found when testing the null hypothesis, odds ratio = 1.0 in logistic regression models for the product term (log of *Ae. aegypti* abundance) against rainfall anomaly x LST anomaly.

**Table 4. Logistic Analysis of Anomaly Classification and *Ae. aegypti* Abundance.**

| Outcome | Climate Classification | Coefficient | Standard Error | Adjusted OR (95% CI) | p-Value |
|---|---|---|---|---|---|
| **Low *Ae. aegypti* Abundance (Ovitrap)** | *Drought* | 0.38 | 0.43 | 1.46 (0.62, 3.42) | 0.39 |
| | *Flood* | 0.41 | 0.25 | 1.51 (0.92, 2.46) | 0.10 |
| | *Heat Wave* | -1.13 | 0.17 | 0.32 (0.23, 0.44) | **<0.001** |
| | *Cold Wave* | -1.37 | 0.11 | 0.25 (0.20, 0.32) | **<0.001** |
| **High *Ae. aegypti* Abundance (Ovitrap)** | *Drought* | -0.36 | 0.13 | 0.70 (0.54, 0.90) | **0.01** |
| | *Flood* | 2.62 | 0.38 | 13.8 (6.5, 29.3) | **<0.001** |
| | *Heat Wave* | -1.53 | 0.03 | 0.22 (0.20, 0.23) | **<0.001** |
| | *Cold Wave* | 0.05 | 0.19 | 1.05 (0.73, 1.52) | 0.78 |
| **Low *Ae. aegypti* Abundance (Pupal)** | *Drought* | 0.88 | 0.40 | 2.41 (1.09, 5.32) | **0.03** |
| | *Flood* | -0.74 | 0.39 | 0.48 (0.22, 1.03) | 0.06 |
| | *Heat Wave* | 0.80 | 0.39 | 2.22 (1.04, 4.75) | **0.04** |
| | *Cold Wave* | 0.28 | 0.27 | 1.32 (0.77, 2.25) | 0.30 |
| **High *Ae. aegypti* Abundance (Pupal)** | *Drought* | -0.16 | 0.06 | 0.85 (0.77, 0.95) | **0.004** |
| | *Flood* | -0.14 | 0.26 | 0.87 (0.53, 1.44) | 0.59 |
| | *Heat Wave* | -0.07 | 0.07 | 0.93 (0.81, 1.07) | 0.30 |
| | *Cold Wave* | 0.84 | 0.23 | 2.33 (1.48, 3.64) | **<0.001** |
| **Low *Ae. aegypti* Abundance (Prokopack)** | *Drought* | 0.50 | 0.63 | 1.65 (0.48, 5.69) | 0.43 |
| | *Flood* | 0.58 | 0.58 | 1.79 (0.57, 5.64) | 0.32 |
| | *Heat Wave* | 0.40 | 0.63 | 1.49 (0.43, 5.13) | 0.53 |
| | *Cold Wave* | -0.56 | 0.12 | 0.57 (0.45, 0.72) | **<0.001** |
| **High *Ae. aegypti* Abundance (Prokopack)** | *Drought* | -1.59 | 0.09 | 0.20 (0.17, 0.24) | **<0.001** |
| | *Flood* | -0.10 | 0.47 | 0.91 (0.36, 2.26) | 0.83 |
| | *Heat Wave* | -1.11 | 0.28 | 0.33 (0.19, 0.57) | **<0.001** |
| | *Cold Wave* | 0.89 | 0.57 | 2.43 (0.80, 7.39) | 0.12 |
| **Low *Ae. aegypti* Abundance (BG-Trap)** | *Drought* | -0.13 | 0.44 | 0.87 (0.30, 2.07) | 0.76 |
| | *Flood* | -0.13 | 0.62 | 0.88 (0.26, 2.93) | 0.83 |
| | *Heat Wave* | 0.11 | 0.25 | 1.12 (0.68, 1.85) | 0.66 |
| | *Cold Wave* | -0.14 | 0.10 | 0.87 (0.71, 1.06) | 0.17 |
| **High *Ae. aegypti* Abundance (BG-Trap)** | *Drought* | 0.54 | 0.40 | 1.71 (0.78, 3.75) | 0.18 |
| | *Flood* | 0.88 | 0.29 | 2.41 (1.36, 4.27) | **0.002** |
| | *Heat Wave* | -0.57 | 0.52 | 0.67 (0.20, 1.58) | 0.28 |
| | *Cold Wave* | 0.26 | 0.65 | 1.30 (0.36, 4.66) | 0.69 |

Multinomial logistic regression adjusted for site, month, and year, with each color representing a different trapping method. Reference category is intermediate *Ae. aegypti* abundance.

***Ae. aegypti* pupae.** Our generalized log-linear model did not find any significant effects of rainfall anomalies, LST anomalies, and the interaction between rainfall anomalies and LST anomalies on pupal abundance.

Similarly, for *Ae. aegypti* pupae, we observed that lower-than-expected levels of rainfall, as are consistent with the definition of a drought, promoted lower *Ae. aegypti* pupal abundance (OR = 2.41 [1.09, 5.32], p = 0.03) and significantly lowered the odds of high pupal abundance (OR = 0.85 [0.77, 00.95], p = 0.004) (Table 4). Heat waves significantly increased the odds of low *Ae. aegypti* pupal abundance (OR = 2.22 [1.04, 4.75], p = 0.04) while cold waves significantly increased the odds of high *Ae. aegypti* pupal abundance (OR = 2.33 [1.48, 3.64], p < 0.001) (Table 4).

***Ae. aegypti* adults.** For adult *Ae. aegypti* abundance collected with Prokopack aspirators, the severity of LST anomalies significantly influenced vector abundance: for every 1°C increase in a monthly LST anomaly, an 8.7% decrease in *Ae. aegypti* abundance would be expected the following month (Table 3). However, for adult *Ae. aegypti* collected with BG-traps, we did not observe a significant relationship between anomaly severity and vector abundance (Table 3).

**Table 5.  Anomaly Severity and Dengue Transmission.**

| Variable | Coefficient | Standard Error | z-Value | Adjusted OR (95% CI) | p-Value |
|---|---|---|---|---|---|
| Rainfall Anomaly | 2E-3 | 4E-3 | 0.55 | 1.00 (0.99, 1.01) | 0.58 |
| LST Anomaly | -0.12 | 0.12 | -1.02 | 0.89 (0.69, 1.11) | 0.31 |

Binomial regression model adjusted for site, month, and year. Test of interaction: P value of 0.88 found when testing the null hypothesis, odds ratio = 1.0 in logistic regression models for the product term (intermediate versus high monthly confirmed dengue cases) against rainfall anomaly x LST anomaly.

Rainfall anomalies and the interaction between rainfall anomalies and LST anomalies remained insignificant for both trapping methods. Compared to Chulaimbo, a significantly increased odds of higher adult abundance as collected by Prokopack aspirators was observed in the model for the urban sites of Kisumu (OR = 6.99 [2.87, 17.00], p < 0.001) and Ukunda (OR = 4.21 [1.71, 10.37], p = 0.002) while a decreased odds of higher adult abundance was observed for Msambweni (OR = 0.12 [0.05, 0.26], p < 0.001). A similar relationship was observed for adults collected by BG-traps. Compared to Chulaimbo, a significantly increased odds of higher adult abundance was observed for Kisumu (OR = 18.17 [6.21, 53.44], p < 0.001) while a decreased odds of higher abundance was observed for Msambweni (OR = 0.17 [0.06, 0.43], p < 0.001).

For adult *Ae. aegypti* collected with Prokopack aspirators, heat waves reduced the odds of "high" mosquito abundance by 67% (OR = 0.33 [0.19, 0.57], p <0.001). Similarly, drought reduced the odds of "high" *Ae. aegypti* abundance by 80% (OR = 0.20 [0.17, 0.24], p < 0.001) (Table 4). However, unlike the results from our bivariate analyses (S2 Fig), once site, month, year, and rainfall anomalies were accounted for, cold waves did not significantly increase the odds of "high" vector abundance for Prokopack (OR = 2.43 [0.80, 7.39], p = 0.12). Moreover, cold waves were expected to decrease the odds of "low" vector abundance (OR = 0.57 [0.45, 0.72], p < 0.001), suggesting that abnormally cold temperatures promoted intermediate vector counts (Table 4). For adult mosquitoes as recorded by BG-traps, we observed that floods significantly increased the odds of "high" vector abundance (OR = 2.41 [1.36, 4.27], p = 0.002) when the analysis was controlled for site, month, year, and LST anomaly (Table 4).

## Dengue transmission

Our binomial logistic regression results suggest that both rainfall and LST anomaly severity are not significantly associated with the transmission of dengue in Kenya when using a two-month lag (Table 5). Moreover, site, month, and year did not significantly influence the results, and the interaction between rainfall and LST anomalies was insignificant at p < 0.05. Similarly, when studying the effect that classification of rainfall and LST events has on dengue transmission, we failed to observe any significant effect (Table 6).

**Table 6.  Binary Analysis of Anomaly Classification and Dengue Transmission.**

| Outcome | Variable | Coefficient | Standard Error | Adjusted OR (95% CI) | p-Value |
|---|---|---|---|---|---|
| Higher Dengue Infection Counts | Rainfall Classification | | | | |
| | Drought | -0.63 | 0.46 | 0.53 (0.22, 1.31) | 0.17 |
| | Flood | 0.57 | 0.69 | 1.76 (0.45, 6.88) | 0.41 |
| | LST Classification | | | | |
| | Heat Wave | 0.17 | 0.63 | 1.18 (0.35, 4.04) | 0.79 |
| | Cold Wave | 0.17 | 0.71 | 1.18 (0.29, 4.76) | 0.82 |

Binomial logistic regression adjusted for site, month, and year. Reference category is <7 monthly confirmed dengue infection.

## Discussion

The impact of recent climate extremes on mosquito vector abundance in Kenya is dependent on various factors, and observed results vary depending on the trapping method. Such a finding suggests that the effect of climate extremes differs based on the particular life stage of *Ae. aegypti.*

We found that one month following floods resulted in a significantly greater abundance of *Ae. aegypti* eggs. When adjusting for the potential modifying effects of site, month, and year, we observed flooding to result in a significant increase in the odds of higher egg abundance (Table 4). Moreover, more extreme rainfall resulted in increased egg abundance (Table 3). As suggested by significant differences in the effect of flooding between sites, urbanization likely influences the impact that flooding has on the abundance of *Ae. aegypti* eggs (S1A and S2 Figs). In our stratified bivariate analyses, we observe only rural sites to experience significantly higher egg abundance following flooding (S2 Fig). Rural areas tend to absorb excess water during periods of extreme rainfall better than urban areas and their human-made water catchments [33], potentially preventing a "flooding out" effect and instead providing more stable pools of water for mosquito breeding. Modifying human behavior during floods, such as the use of artificial and human-made containers, outdoor trash disposal, are necessary in removing potential breeding sites for mosquito vectors.

For *Ae. aegypti* pupae, we observed significant increases in abundance following cold waves, but both droughts and heat waves were associated with significantly fewer vectors (Table 4). From the unstratified bivariate analysis, pupal abundance was not significantly associated with extreme climate events (S1B Fig). However, in the stratified analysis, pupal abundance was significantly lower following a heat wave in Chulaimbo, a rural study site in western Kenya (S3B and S4 Figs). Even sites that are close in proximity to one another can experience different climate anomalies due to differences in local topography, wind, etc. Of all trapping methods assessed, pupal counts were consistently the lowest across study sites and were zero for more than a third of the observation periods, potentially affecting this study's ability to assess the effect of extreme climate events on *Ae. aegypti* pupal abundance.

Drier climatic conditions, including drought and heat waves, seem to promote lower vector abundance. In terms of anomaly severity, we observed that more anomalous cold temperatures promoted greater adult abundance, as recorded by Prokopack (Table 3). Our bivariate analysis supports such a finding, as we observed an increased abundance of adult *Ae. aegypti* mosquitoes following periods defined as cold waves, for both trapping methods of Prokopack and BG-traps (S3C and S4C Figs). However, this association was likely driven by the fact that the average observed LST during cold waves was 29.99°C, which is in line with the 29°C optimal ambient air temperature threshold for *Ae. aegypti* [23]. Moreover, site and year were significant predictors of abundance classification, as supported by the multinomial regression and bivariate analyses. For example, many of the of the study's cold wave observations actually took place during the 2015–2016 summer months (S5 Fig). This finding suggests that *Ae. aegypti* find cooler than expected months to be more favorable for growth because of the extremely warm average Kenyan temperatures year-round. However, the most notable finding was that for adult *Ae. aegypti* mosquitoes, as recorded by BG-trap, we observed significantly greater odds of higher adult abundance following floods, which is consistent with what we observed for *Ae. aegypti* eggs (Table 4). Such a finding suggests that of all extreme climate events, flooding most likely contributes to higher adult abundance by providing additional breeding sites for *Ae. aegypti.*

Our results primarily suggest that flood seasons contribute to significantly higher *Ae. aegypti* egg and adult abundance after one month. While such a finding is unique for Kenya, it

is consistent with several previous epidemiological studies [14–17]. Periods defined as "floods" in Kenya considered a monthly accumulation of rainfall that was extreme but that did not always cause physical floodwaters that might have resulted in mosquito habitats being washed away. This was in contrast to previous tightly controlled simulations and experiments, which resulted in a "flooding-out effect" of vector breeding sites [10–13].

When adjusting for site, month, and year, we did not observe statistically significant relationships between extreme climate anomalies (specifically floods and cold waves) and confirmed cases of dengue infection, but this result may still inform our understanding of these relationships. This finding suggests that mosquito abundance and dengue risk do not necessarily share a linear association, as human behaviors can modify the relationship and influence infection risk. For example, previous studies of dengue dynamics in China have observed nonlinear statistical relationships between vectors, human incidence, and climate [29, 34]. This nonlinear relationship is in part driven by the extrinsic incubation period (EIP), the time it takes for the virus to disseminate in the mosquito, and mosquito lifespan. EIP duration and the mosquito lifespan are temperature-dependent. EIP becomes faster at higher temperatures, but the mosquito must survive longer than the EIP for transmission to occur [35]. The 29° C thermal optima derived from the full suite of *Ae. aegypti* temperature-dependent traits and widespread occurrence of the *Ae. aegypti* vector throughout Africa, suggests that as temperatures increase, dengue burden is also likely to increase throughout sub-Saharan Africa [36].

Preventative measures taken by our study population in the four villages may have contributed to reductions in the risk of dengue transmission. A recent study found that the primary control method utilized by participants in our study sites was bed-nets; of 5,833 enrolled patients at our four Kenyan clinic sites, 4,397 (75.4%) reported always using bed-nets [37]. While bed-nets can be protective against several mosquito species, they are not as useful against a diurnal mosquito species like *Ae. aegypti*. In addition to the removal of outdoor breeding sites, greater education and promotion of other preventative measures is necessary. The use of outdoor spray and repellant targeting this exophagic and anthropophagic mosquito species can significantly drive down the increased risk of dengue infection following extreme climate events.

An improved understanding of the relationship between extreme climate and dengue transmission can also allow for the development of climate-based early warning systems in Kenya. The El Niño Southern Oscillation (ENSO) and Indian Ocean Dipole (IOD) are responsible for changes in the sea-surface temperature that moderate seasonal rainfall and temperature variability in eastern Africa. As such, there is an increased likelihood of extreme climate events during extreme ENSO and positive IOD. By discerning the spatiotemporal relationship between extreme climate and dengue transmission, we can use ENSO and IOD indices to predict periods of heightened arboviral infection risk.

This study effectively investigates the impact of recent climate extremes on various life stages of *Ae. aegypti* abundance. With the availability of long-term satellite data, monthly rainfall and LST anomalies were calculated in order to determine associations with vector abundance and disease risk. A strength of the study is the calculation of monthly anomalies to explain both mosquito and disease data trends. The observational nature of the study offers high external validity. The study's results can be generalized to dengue endemic regions similar in climate and demography to Kenya and the effects related to rainfall and LST can be more widely applicable. However, there are several limitations that are important to consider for future studies.

Due to the lack of long-term climate data available for analysis, this study makes use of satellite-derived data for four study sites in Kenya, which can provide us with a reference for the effect of extreme climate events more generally in the region (S6 Fig). As such, satellite-derived

climate data offers the ability to compare deviations in recent climate conditions from long-term records. Minimum and maximum temperature values can impact temperature variability to differing extents. We are unable at the moment to reconstruct the data as produced and provided to tease out the effect of minimum and maximum temperatures. so future studies should consider temperature anomalies in the context of these extreme values rather than the mean. Additionally, future studies can make use of lag periods of differing temporal resolutions. Our lag periods were established based on the traditional cycle of dengue infection [29–31]; however, it is not a perfect model and other lag periods, especially with regards to dengue risk, have been observed in the literature. Furthermore, it is necessary to consider "anomalies" in the context of established biologically relevant thresholds for both temperature and rainfall.

Underreporting and misclassification of fever can influence results as well. Since DENV is heavily underreported in the region, all enrolled participants were tested for dengue and malaria. In a study of undifferentiated fever in Kenya between 2014–2017, 150 (51.5%) of 291 participants with dengue viremia were malaria smear positive, suggesting a large overlap in infections [38]. Additionally, flaviviruses, which include West Nile virus (WNV), ZIKA virus (ZIKAV), and yellow fever virus (YFV), have the potential to cross-react with DENV. However, WNV has a comparatively lower seroprevalence compared to DENV [26] while ZIKAV and YFV are essentially absent from our study sites.

As climate change accelerates and increases the intensity and severity of extreme climate events, understanding how they impact infectious disease transmission is essential. Climate change continues to blur and transform previously discrete seasons, which will influence vector dynamics and disease burden in Kenya. Ultimately, efforts should be focused on eliminating mosquito-breeding sites through the removal of human-made containers and trash in order to reduce vector populations. At the same time, encouragement of the use of spray, coils, and repellant can reduce the heightened risk of viral transmission during periods of anomalous climate and more specifically, floods.

## Supporting information

**S1 Appendix. Land Surface Temperature Data.**
(DOCX)

**S2 Appendix. Mosquito Trapping Method Timing and Frequency.**
(DOCX)

**S3 Appendix. Crude Bivariate Analysis Methods.**
(DOCX)

**S1 Fig. Effect of Rainfall Anomalies on Vector Abundance.** Boxplot of vector abundance by A) ovitrap B) Prokopack C) BG-trap and D) pupal trap one month following anomaly classified as normal, drought, or flood. Wilcoxon test p-values displayed between groups.
(TIF)

**S2 Fig. Effect of Rainfall Anomalies on Egg Abundance Stratified by Urbanization.** Boxplot of Ae. aegypti egg abundance one month following anomaly classified as normal, drought or flood. Stratified by rural (Chulaimbo and Msambweni) and urban (Kisumu and Ukunda) sites. Wilcoxon test p-values displayed between groups.
(TIF)

**S3 Fig. Effect of LST Anomalies on Vector Abundance.** Boxplot of vector abundance by A) ovitrap B) Prokopack C) BG-trap and D) pupal trap one month following anomaly classified

as normal, heat wave, or cold wave. Wilcoxon test p-values displayed between groups.
(TIF)

**S4 Fig. Effect of LST Anomalies on Adult Abundance Stratified by Geography.** Boxplot of adult *Ae. aegypti* abundance (Prokopack) one month following anomaly classified as normal, heat wave or cold wave. Stratified by western (Kisumu and Chulaimbo) and coastal (Ukunda and Msambweni) sites. Wilcoxon test p-values displayed between groups.
(TIF)

**S5 Fig. Anomaly Severity by Site.** Heat map displaying monthly anomaly severity for A) accumulated rainfall and B) average LST between November 2013—February 2019, stratified by study sites.
(TIF)

**S6 Fig. Relationship Between Land Surface Temperature and Ambient Air Temperatures.** Across all four study sites, there is a strong correlation ($R > 0.50$, $p < 0.05$) between land surface temperature and ambient air temperatures between November 2013 –February 2019; however, there is clear variability between the two measurements.
(TIF)

**S1 Table. Characteristics of Rainfall Classification.** Mean and standard deviation of monthly LST (˚C), LST anomaly (˚C), monthly rainfall (mm), rainfall anomaly (mm), monthly ambient air temperature (˚C), and monthly humidity (%) between previously defined groups of "flood," "drought," and "normal rainfall." LST and rainfall anomalies refer to difference between observed monthly values and long-term means. p-values indicate significance values from Kruskal-Wallis Rank Sum tests among three groups. *$p \leq 0.05$, ** $p \leq 0.01$, *** $p \leq 0.001$ for Wilcoxon-test where "Normal Rainfall" is considered the reference group.
(DOCX)

**S2 Table. Characteristics of LST Classification.** Mean and standard deviation of monthly LST (˚C), LST anomaly (˚C), monthly rainfall (mm), rainfall anomaly (mm), monthly ambient air temperature (˚C), and monthly humidity (%) between previously defined groups of "heat wave," "cold wave," and "normal LST." LST and rainfall anomalies refer to difference between observed monthly values and long-term means. p-values indicate significance values from Kruskal-Wallis Rank Sum tests among three groups. Note: *$p \leq 0.05$, ** $p \leq 0.01$, *** $p \leq 0.001$ for Wilcoxon-test where "Normal LST" is considered the reference group.
(DOCX)

**S3 Table. Outcome Classification Cutoffs.**
(DOCX)

**S4 Table. Missing Data for Outcomes.** Missing data was excluded from the regression models.
(DOCX)

## Acknowledgments

This retrospective study assesses the effect of extreme climate events on mosquito abundance and DENV transmission in Kenya using data from a study performed by the LaBeaud Lab, Technical University of Mombasa, and the Kenya Medical Research Institute (KEMRI). Vector, climate, and disease data is housed in the Research Electronic Data Capture Database (REDCap) at Stanford University. The Stanford REDCap platform (http://redcap.stanford.edu) is operated by Stanford Medicine Research IT team.

## Author Contributions

**Conceptualization:** Cameron Nosrat, Jonathan Altamirano, Assaf Anyamba, Jamie M. Caldwell, A. Desiree LaBeaud.

**Data curation:** Cameron Nosrat, Assaf Anyamba, Jamie M. Caldwell, Richard Damoah.

**Formal analysis:** Cameron Nosrat, Jamie M. Caldwell.

**Funding acquisition:** A. Desiree LaBeaud.

**Investigation:** Cameron Nosrat, Assaf Anyamba, Jamie M. Caldwell.

**Methodology:** Cameron Nosrat, Jonathan Altamirano, Assaf Anyamba, A. Desiree LaBeaud.

**Project administration:** Francis Mutuku, Bryson Ndenga, A. Desiree LaBeaud.

**Resources:** Francis Mutuku, Bryson Ndenga, A. Desiree LaBeaud.

**Software:** Cameron Nosrat, Jonathan Altamirano, Jamie M. Caldwell.

**Supervision:** Francis Mutuku, Bryson Ndenga, A. Desiree LaBeaud.

**Validation:** Cameron Nosrat, Jonathan Altamirano, Assaf Anyamba, Jamie M. Caldwell.

**Visualization:** Cameron Nosrat.

**Writing – original draft:** Cameron Nosrat.

**Writing – review & editing:** Cameron Nosrat, Jonathan Altamirano, Assaf Anyamba, Jamie M. Caldwell, Richard Damoah, Francis Mutuku, Bryson Ndenga, A. Desiree LaBeaud.

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
