## [Decision Letter · Decision Letter 0]

20 Sep 2020

Dear Mr. Nosrat,

Thank you very much for submitting your manuscript "Impact of Recent Climate Extremes on Mosquito-Borne Disease Transmission in Kenya" for consideration at PLOS Neglected Tropical Diseases. As with all papers reviewed by the journal, your manuscript was reviewed by members of the editorial board and by several independent reviewers. In light of the reviews (below this email), we would like to invite the resubmission of a significantly-revised version that takes into account the reviewers' comments. 

We cannot make any decision about publication until we have seen the revised manuscript and your response to the reviewers' comments. Your revised manuscript is also likely to be sent to reviewers for further evaluation.

Sincerely,

Elvina Viennet, PhD

Deputy Editor

Elvina Viennet

Deputy Editor

Reviewer's Responses to Questions

**Key Review Criteria Required for Acceptance?**

**Methods**

-Are the objectives of the study clearly articulated with a clear testable hypothesis stated?

-Is the study design appropriate to address the stated objectives?

-Is the population clearly described and appropriate for the hypothesis being tested?

-Is the sample size sufficient to ensure adequate power to address the hypothesis being tested?

-Were correct statistical analysis used to support conclusions?

-Are there concerns about ethical or regulatory requirements being met?

Reviewer #1: No

Reviewer #2: The paper is addressing an important component of climate change in relation to an arbovirus infections. The effects of climate extremes on vector-born diseases is not well understood. Attempts have been made to use statistical, dynamic and process based model to explain the interaction of the various meteorological variables and their final impact on vectors and pathogens. The complexity of a model does not0 allways provide the expected results. A clear understanding of the biology of transmission in relationship to meteorological variables is critical in modelling transmission.

In most cases the relationship between temperature and the viral extrinsic incubation period is often non-linear. Lags of unknown periods are also required for the breeding habitats to stabilize. Some extreme rainfall and temperature are lethal vectors and the virus. In other cases rainfall is the only controlling factor as the temperature conditions are close to ideal for hypertransmission.

The strength of this paper lies in its exploratory aproach in finding the relationship between extreme climate events and dengue transmission. However it suffers from the use of satellite derived data. Among the shortfall of this data is failure to disentangle the impacts of variability in the minimum and maximum temperature. In some sites over 95% of the temperature variability comes from the maximum temperature while in some sites the minimum temperature may contribute as much as 20% of the anomalies. Thus while the mean temperature may fail to reveal the full extent of anomalies, the max and min temperatures could be telling a different story between sites.

Temperature and rainfall threasholds define the windows of transmission and it is importsnt to identify these threasholds while developinng statistical models. Care must be taken in usind statistically defined extreme meteorological conditions without making reference to the trsnsmission windows.

Reviewer #3: The objectives of the study, identify how periods of extreme rainfall and temperature affect mosquito abundance and the risk of dengue infection, is clearly articulated, but not formulated in the form of a hypothesis. The study design appropriate to address the stated objectives, the data set is clearly described.

Monthly mean land surface temperatures was gridded at 0.05° while rainfall at 0.1°. Do you expect any scale dependency in the results comparing temperature with rainfall effects? The measurement of the vector abundance using different types of traps is well thought out.

**Results**

-Does the analysis presented match the analysis plan?

-Are the results clearly and completely presented?

-Are the figures (Tables, Images) of sufficient quality for clarity?

Reviewer #1: Yes

Reviewer #2: The analysis seems adequate to adress the experimental design. The ststistical tests seem appropriet for the types data presented. 

The tables are well preseed and so are the other illustrations.

In table 1, colum 3 data does not add up to 204. It adds up to 194. Please check and correct

Reviewer #3: The results are clearly and completely presented, very informative details are given in the supplemental. Figures and tables are of very good quality for clarity. Some suggestions should be taken into account:

L261: Table 1. Distribution of Rainfall Categorization: you may use different background color or grey to divide between sites prone to drought events (Chulaimbo, Kisumu) and those sites prone to flood events (Msambweni, Ukunda). To better understand this result, it would be helpful to support the reader with a map of the elevation of Kenia, e.g. within the study site map.

L274: "By modeling such relationships, we hoped to control for ..." In the text, it remains unclear what your results show. The sentence implicit asks for "...., but....". This will follow only later in L282. Please rephrase your paragraph introduction.

L299: Even if there is a repetition in the wording, I suggest to keep the word "egg abundance" instead of vector abundance.

L301ff: Doubling the information about OR in text while it is shown already in table 4 can be questioned.

**Conclusions**

-Are the conclusions supported by the data presented?

-Are the limitations of analysis clearly described?

-Do the authors discuss how these data can be helpful to advance our understanding of the topic under study?

-Is public health relevance addressed?

Reviewer #1: No

Reviewer #2: The conclusions are supported by the analysis which is sufficiently rigorous. This being a review paper, the authors should discusd the effects of temperature on dengue virus extrinsic incubation period (EIP). In the case of malaria rainfall has a near linear relationship with vector abundance, within 3-4 months within the rainy season, Threaftet ecological changes disruptt this relationship. However imperical evidence indicates that temperature has an exponential effect on the EIP even when the small temperature changes occurring at a critical transmission threashold may not be statistically significant. 

The authors have been explicit on the limitations of their study and in particular the use of satellite data. Aquisation is station data can be an expensive undertaking. However this problem can be resolved by collaborating with the Kenya Meorological Department. Normally they operate all weather and climate based early warning systems and are there for key stakeholders in these types of studies.

The authors recognize that thier research could lead to the development of an weather based early warning system. It would be usefull to discuss the effects is ENSO and IOD on extreme weather in Kenya and East Africa.

Reviewer #3: The conclusions are supported by the presented data and the limitations are mainly discussed (L439ff). Please discus possible problems in PCR or Elisa which can occur transporting the blood from Kenia to Standford and which cross reactions may occur? Was capacity building taken into consideration to do the blood measures in Kenia? Future research is outlined, and the public health relevance is addressed (L464ff).

L374-384: The terms urbanization and rural areas are only introduced in the discussion. In this paragraph you bring up new results and discus them. I would have expect these additional results which are shown in the supplemental already described at the end of the results chapter "Ae. aegypti eggs". 

L411: Is an adaptation of Ae. aegypti towards warmer temperatures expected in the literature?

L422: "...that mosquito abundance and dengue risk do not necessarily share a linear association..". Please compare with further literature from other regions (e.g. Ruiyun 2018 PNAS). 

L432: Does underreporting or misclassification of fever can play a role, too? This is often described in other studies (e.g. WHO 2019, update of the dengue situation).

L432ff: Besides indirect effects of temperature on mosquito abundance, direct effects of temperature on the transmission should be mentioned here, too (e.g. temperature depend extrinsic incubation period of dengue, reviewed in Tjaden 2013 PlosNTD).

**Editorial and Data Presentation Modifications?**

Reviewer #1: (No Response)

Reviewer #2: Please check additions in table 1 column 3.

Reviewer #3: L107: "... higher monthly rainfall up to 16.2mm over the past decade." The absolute value of mm increase within decades, does not help to understand if this is an (nearly) extrem value or not. Maybe you add the percentage of the increase.

L151: "0.05° MOD11C3 " you may add the km as done for rainfall "gridded at 0.1° x 0.1° (≈11 km)".

L224: Please change Aedes aegypti in Ae. aegypti.

L258: "There is no standard for such designations, ", you can justify your approach by citing IPCC 2012.

L261: Table 1. Distribution of Rainfall Categorization: you may use different background color or grey to divide between sites prone to drought events (Chulaimbo, Kisumu) and those sites prone to flood events (Msambweni, Ukunda). To better understand this result, it would be helpful to support the reader with a map of the elevation of Kenia, e.g. within the study site map.

L344: Change table 5 to table 4.

L395: Do you mean "transmission by Ae. aeypti..." or "transmission for dengue"?

**Summary and General Comments**

Reviewer #1: (No Response)

Reviewer #2: Climate change is having and impact on the geographic range, seasonalty and the evolition of vector-borne diseases (VBD) epidemics. Among the adaptation options is the ability to predict these trends and to develop early warning systems to enable pre-emptive interventions. Extreme weathet events are often associated with VBD epidemics. It is criticslly important to understanf the biology of trandmission in relationship to the extremr weathet events. This paper highlights efforts towards that direction. As an exploratory study It has indicated how available entomological, virological and meteorological data analysis may lead to the understanding of extreme weather events on dengue fever.

The weakness of the paper lies in the failure to include critical and well known temperature threasholds of dengue virus transmission biology. Equaly rainfal has lower and upper threasholds that either support vector breeding or disrupt it. Thus statisticall defined extremes must take cognisance of biologically relevant thresholds.

Although the paper has discussed the stelite data limitations it shout highlight these other limitation so that they are avoided in futurr work.

The research has no outstanding ethical issues.

The paper should address these concerns without undertaking new data analysis.

Reviewer #3: This study is topically and of high interest. The authors aim to close a knowledge gap on the effects on extreme events on mosquito-borne diseases. The results can be helpful to consider mosquito control actions and public health interventions in time. 

in my opinion a short introduction in the wording with regard to climate change is missing, including trends, climate variability and extreme events (e.g. extreme events following natural internal climate variability (which is the element of climate variability that arises spontaneously within the climate system even in the absence of forcings, see IPCC AR5 2014) or due to climate warming). Also it would be helpful, if you can refine your wording with regard to "extreme weather events" and "extreme climate" (see IPCC 2012, Managing the risk of extreme events, 3.1.2.).

PLOS authors have the option to publish the peer review history of their article (what does this mean?). If published, this will include your full peer review and any attached files.

Reviewer #1: No

Reviewer #2: Yes: Dr. Andrew Karanja Githeko. PhD

Reviewer #3: No
---

## [Decision Letter · Decision Letter 1]

26 Jan 2021

Dear Mr. Nosrat,

We are pleased to inform you that your manuscript 'Impact of Recent Climate Extremes on Mosquito-Borne Disease Transmission in Kenya' has been provisionally accepted for publication in PLOS Neglected Tropical Diseases.

Best regards,

Elvina Viennet, PhD

Deputy Editor

Elvina Viennet

Deputy Editor

Reviewer's Responses to Questions

**Key Review Criteria Required for Acceptance?**

**Methods**

-Are the objectives of the study clearly articulated with a clear testable hypothesis stated?

-Is the study design appropriate to address the stated objectives?

-Is the population clearly described and appropriate for the hypothesis being tested?

-Is the sample size sufficient to ensure adequate power to address the hypothesis being tested?

-Were correct statistical analysis used to support conclusions?

-Are there concerns about ethical or regulatory requirements being met?

Reviewer #2: None

Reviewer #3: Yes

**Results**

-Does the analysis presented match the analysis plan?

-Are the results clearly and completely presented?

-Are the figures (Tables, Images) of sufficient quality for clarity?

Reviewer #2: The analysis is fine and appropreate. The results are clearly presented in tables and figures..

Reviewer #3: yes

**Conclusions**

-Are the conclusions supported by the data presented?

-Are the limitations of analysis clearly described?

-Do the authors discuss how these data can be helpful to advance our understanding of the topic under study?

-Is public health relevance addressed?

Reviewer #2: The conlusions are supported by the results and their limitations have been clearly stated. The authors have indicated how their findings have advanced the understanding of the phenomenon they were studying. The paper has indicated how the findings can contribute towards the development of an early warning system for climate driven dengue epidemics.

Reviewer #3: yes

**Editorial and Data Presentation Modifications?**

Reviewer #2: Not required in this reviced manuscript.

Reviewer #3: accept

**Summary and General Comments**

Reviewer #2: The overall clarity of the paper has improved in this version and the authors have justified why they used satellite derived data rather than station data.

Reviewer #3: no furhter comments

PLOS authors have the option to publish the peer review history of their article (what does this mean?). If published, this will include your full peer review and any attached files.

Reviewer #2: **Yes: **D. Andrew K. Githeko PhD

Reviewer #3: No

---

## [Editor Report · Acceptance letter]

26 Feb 2021

Dear Mr. Nosrat,

We are delighted to inform you that your manuscript, "Impact of Recent Climate Extremes on Mosquito-Borne Disease Transmission in Kenya," has been formally accepted for publication in PLOS Neglected Tropical Diseases.

Best regards,

Shaden Kamhawi

co-Editor-in-Chief

Paul Brindley

co-Editor-in-Chief
